# communications
## engineering

# A generative force model for surgical skill quantification using sensorised instruments

Artūras Straižys [1], Michael Burke [1,2], Paul M. Brennan[1] & Subramanian Ramamoorthy [1,3✉]

Surgical skill requires the manipulation of soft viscoelastic media. Its measurement through generative models is essential both for accurate quantification of surgical ability and for eventual automation in robotic platforms. Here we describe a sensorised scalpel, along with a generative model to assess surgical skill in elliptical excision, a representative manipulation task. Our approach allows us to capture temporal features via data collection and downstream analysis. We demonstrate that incision forces carry information that is relevant for skill interpretation, but inaccessible via conventional descriptive statistics. We tested our approach on 12 medical students and two practicing surgeons using a tissue phantom mimicking the properties of human skin. We demonstrate that our approach can bring deeper insight into performance analysis than traditional time and motion studies, and help to explain subjective assessor skill ratings. Our technique could be useful in applications spanning forensics, pathology as well as surgical skill quantification.

[1] University of Edinburgh, Edinburgh, UK. [2] Monash University, Clayton, VIC, Australia. [3] Alan Turing Institute, London, UK. ✉email: s.ramamoorthy@ed.ac.uk

Time and motion studies are frequently used to model, analyse and understand complex human manipulation tasks. This remains the case in the context of deformable tissue handling or manipulation, despite broad acknowledgement of the importance and role of forces in these tasks. For the most part, this reliance on kinematic sensing is due to a limited ability to measure forces at the tool-tissue interface. The ability to capture high fidelity information at this interface is key to downstream applications and analysis across a broad range of research areas, including pathology, forensics and surgical skill understanding. In this work, we introduce a low cost, easy-to-replicate tool and accompanying models that enable this.

As an example, this work considers the surgical procedure of elliptical excision, in which skin incisions are made along a parabolic curve. As is the case for many important and practical manipulation tasks, the outcome and the quality of task execution directly depends on both the overall amplitude and the temporal characteristics of the applied forces. Throughout an incision, the non-dominant hand applies continuous tension to the tissues surrounding the cutting contour, while the dominant hand controls the scalpel's movement[1]. Successful tissue dissection implies the application of appropriate force levels[2]—sufficient for deliberate and controlled tissue separation, but not too excessive to avoid iatrogenic tissue damage[3]. In addition, cutting forces are continuously modulated by active tissue tensioning and the scalpel's nonholonomic-like movement through viscoelastic tissues.

Despite the central role that forces play in surgery[4–6], the analysis of these remains a novel area of research[2], as the majority of developed methods for analysing these skills are vision-based and mainly focus on instrument motion[7–10]. However, there is some evidence that force-based performance metrics can be superior to metrics that are based on movement alone[11]. In addition, recent studies indicate that tool-tissue interaction forces can uniquely reflect a surgeon's competence[12]. Interestingly, studies show lack of correlation between tool-tissue forces and motion parameters[13]. Moreover, unlike motion parameters[14,15], force parameters show no correlation with the execution time of surgical tasks[16,17]. The above body of evidence indicates that the force modality may offer distinct information that is largely ignored by time and motion studies.

When force sensing is employed, the descriptive statistics applied by most studies disregard the temporal structure of force measurements under stationarity assumptions. This assumption is highly unrealistic for tasks like elliptical excision, where viscoelastic properties of tissues and a set of distinct phases of task execution cause the forces to exhibit strong time-dependent behaviour (Fig. 1a, b). Here, we propose and use a generative model of elliptical excision forces to encode the behavioural characteristics of the task execution. In our method, we extend the switching dynamics of a Markov model[18,19] with a latent continuous dynamical system that captures the viscoelastic properties of scalpel-tissue interaction[20,21]. Our proposed elliptical excision force model captures the following components of the observed behaviour: 1) the step-like force profile with distinct transient and steady-state phases, 2) the amplitude and envelope of the force profile, characterised by the upper and the lower force boundaries, 3) the variation of the force magnitude in both transient and steady-state phases, and 4) the smoothness of task execution flow, characterised by the frequency of interruptions due to discrete events of tissue re-tensioning or finger re-positioning.

This paper shows that a) these components can compactly describe the execution of elliptical excisions, b) our generative model offers greater insight into analysis of skill when compared to descriptive statistics, and c) the model can quantify the

subjective evaluation of excision skills and enable the comparison of expert assessors with differing implicit assessment criteria. In order to apply this model to investigate scalpel cutting skills[22–24] in an elliptical excision task, we first developed a low-cost sensorised scalpel and an easy-to-replicate multilayered skin-mimicking phantom (Fig. 1c, d). We then collected a dataset of 12 incision force profiles from 12 medical students (Fig. 2), with video recordings of these incisions evaluated by surgical experts (Supplementary Movie), followed by performance analysis using traditional force-based descriptive statistics. Finally, we contrasted this approach with our generative model and found our model superior to descriptive statistics in terms of its ability to analyze the surgical skill and the implicit criteria employed by experts during evaluation.

To summarize, our core findings in this study are as follows:

- Force sensing at the tool-tissue interface enables detailed analysis of manipulation tasks and surgical skill quantification that can be aligned with expert evaluation criteria.
- Commonly considered descriptive statistics that fail to account for non-stationarity are severely limited here, and force-based analysis of manipulation tasks requires a model that explicitly decomposes observations into amplitude and temporal components.

## Results

Figure 2 shows the distribution of force profiles (mean and standard deviation) for each of the 12 medical students (blue), compared with force profiles of two practising surgeons - consultant neurosurgeon (dark yellow) and plastic surgeon (green), each with 5 years of experience. There is a considerable difference in the mean, variability (standard deviation) and overall shape (envelope) of the incision force profiles across the subjects. For example, force profiles of subjects $H$ and $J$ resemble an overdamped step-like response with a smooth and even force level in the steady-state phase of the excision, whereas force profiles of surgeon $A$ (dark yellow) show noticeable force modulation (e.g. dip in the force at $t = 3$ s). The narrow envelope of the profile distribution (i.e. force profile variability) in the surgeon's trials indicate that such modulation is consistent, and hence, is likely to be a part of the cutting behaviour.

**Subjective evaluation of the incision skills**. Four surgical experts (two plastic surgeons and two neurosurgeons) subjectively evaluated all 15 trials (12 original trials plus 3 repeated, see Methods section for details) independently, based on trial videos (Supplementary Movie). The experts were asked to group the trials according to their perceived proficiency (i.e. experts were free to evaluate the performance according to the criteria of their own choice) and provide comments to support their judgement (Supplementary Tables 1–4). Supplementary Fig. 1 shows the boxplots of the grouped subjects based on proficiency rating from 0 to 3 (where 0 is the poorest performance).

The assessment showed poor inter-rated agreement[25] among the experts (Supplementary Fig. 2), with intraclass correlation coefficient (two-way random, single measures) of 0.45.

Despite agreeing in their assessments of the poorest performances (both subject $F$ and the second trial of subject $C$ were rated the worst by each of the experts), experts showed a noticeable difference in rating the average and top performers. For example, Expert A rated subject $G$ with the highest score of 3, while both experts B and C rated it as the second poorest performer (score 1). In addition, subjects $E$ and $H$ were rated with the highest score by experts C and D, but only with a second-lowest score by expert A. Finally, experts A and C rated the first

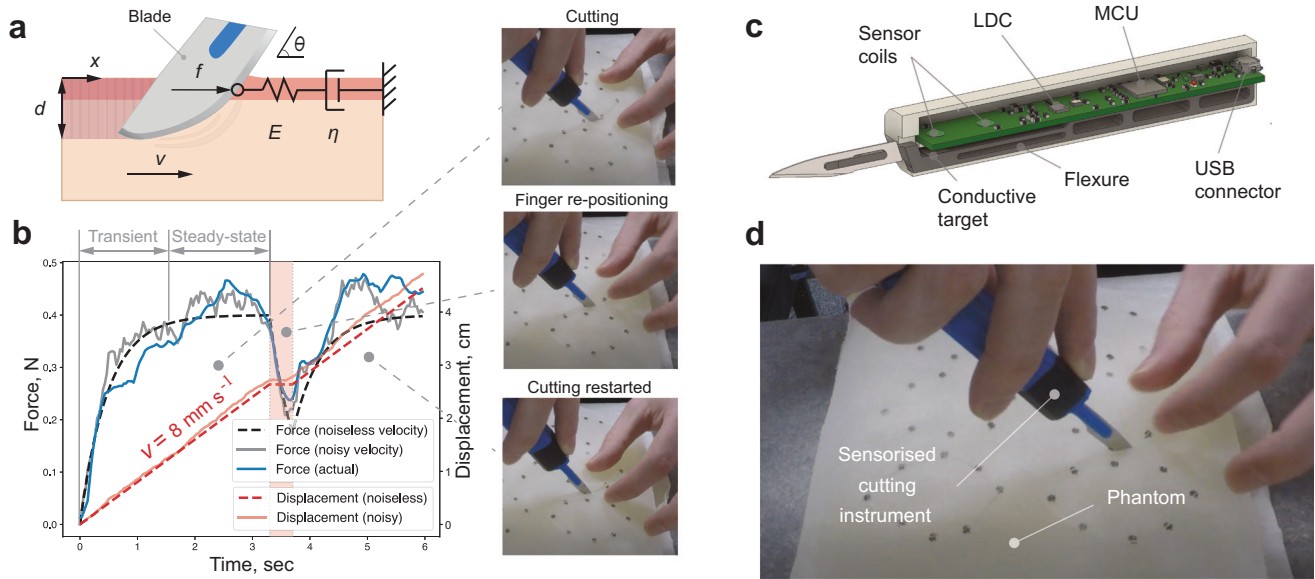

**Fig. 1 Overview of the proposed elliptical excision force model, sensorised scalpel and the experiment. a** Maxwell model of the cutting process, where $x$ denotes blade's displacement, $d$ is depth of excision, $\theta$ is angle of blade insertion, $v$ is blade's velocity, $E$ and $\eta$ are spring and damping coefficients, respectively. (Pink and ivory colours denote the outer and inner layers of tissue phantom, respectively. The shaded area corresponds to the phantom region separated by the blade.) **b** Generated incision force and blade displacement profiles versus the actual incision force measurement (blue). ($E = 1\,\mathrm{N\,cm^{-1}}$, $\eta = 0.5\,\mathrm{N\,s\,cm^{-1}}$, $v \in [0, 8]\,\mathrm{mm\,s^{-1}}$ with standard deviation of $0.35\,\mathrm{mm\,s^{-1}}$). **c** Concept design of the sensorised scalpel (here LDC is Inductance-to-Digital converter, MCU is Micro-controller Unit and USB is Universal Serial Bus). **d** The experiment: 12 medical students and two professional surgeons were asked to perform a series of 12 elliptical excisions on a tissue phantom.

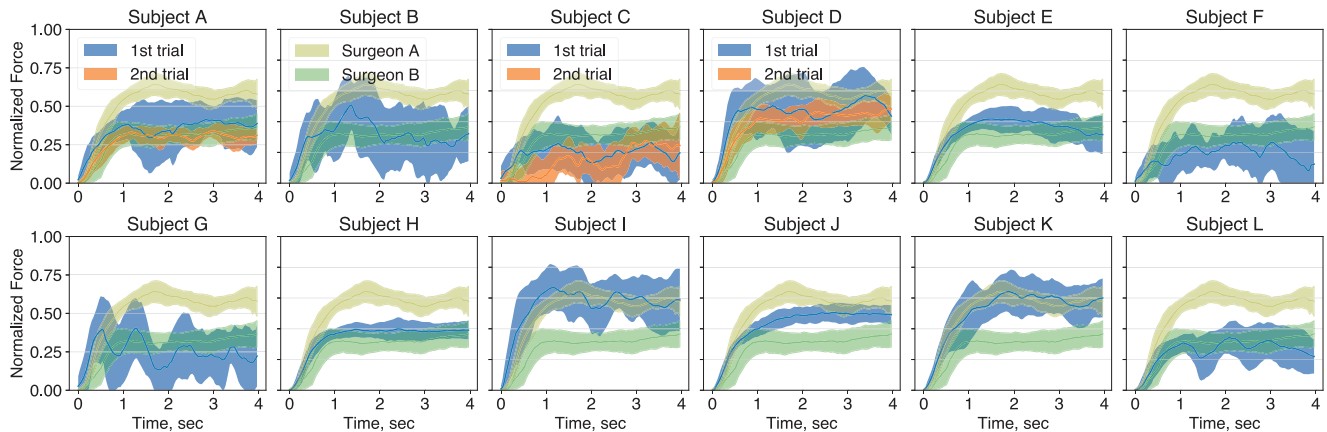

**Fig. 2 Subject-specific distributions of the excision force profiles.** Mean and standard deviation ($N = 12$) of normalized force profiles for each of the medical students (blue) and practicing surgeons (dark yellow and green). Subjects A, C and D have repeated the trials after two months (orange). Supplementary Fig. 4a shows individual force profiles for each subject.

trial of subject $A$ with the highest score, but it was rated as the second poorest by experts B and D.

The discrepancies in subjective assessment of the overall proficiency perceived by the experts highlight the challenges in teaching and assessing skills that are typically mastered through apprenticeship. These differences in assessment might reflect the different specialities, schools or experience levels of the experts. In this study, we treat each expert assessment as an equally valid evaluation.

Below, we investigate the characteristics of elliptical excision performance that drive each expert's perception of skill. Specifically, we study the relationships between the measured incision forces and the subjective assessments of skill based on motion alone. In the following sections, we perform the analysis using the conventional force-based metrics and introduce a generative model for elliptical excision forces that decomposes

force measurements into a set of independent components that uniquely describe the manner of the excision. Finally, we provide an analysis of how these components can explain the subjective criteria employed by each expert.

**Traditional performance analysis**. In this section, we analyzed the relationships between the subjective evaluations by experts and the following objective force-based metrics: mean force, force variability (standard deviation), peak force, scaled force (mean force divided by the peak force value, an indication of force overshoot), derivative of force with respect to time[11] (indication of the aggressiveness) and force integral (indication of cutting energy). Supplementary Fig. 3 shows the relationships.

Levene's test showed that the variance of incision force samples has a statistically significant difference across the subjects

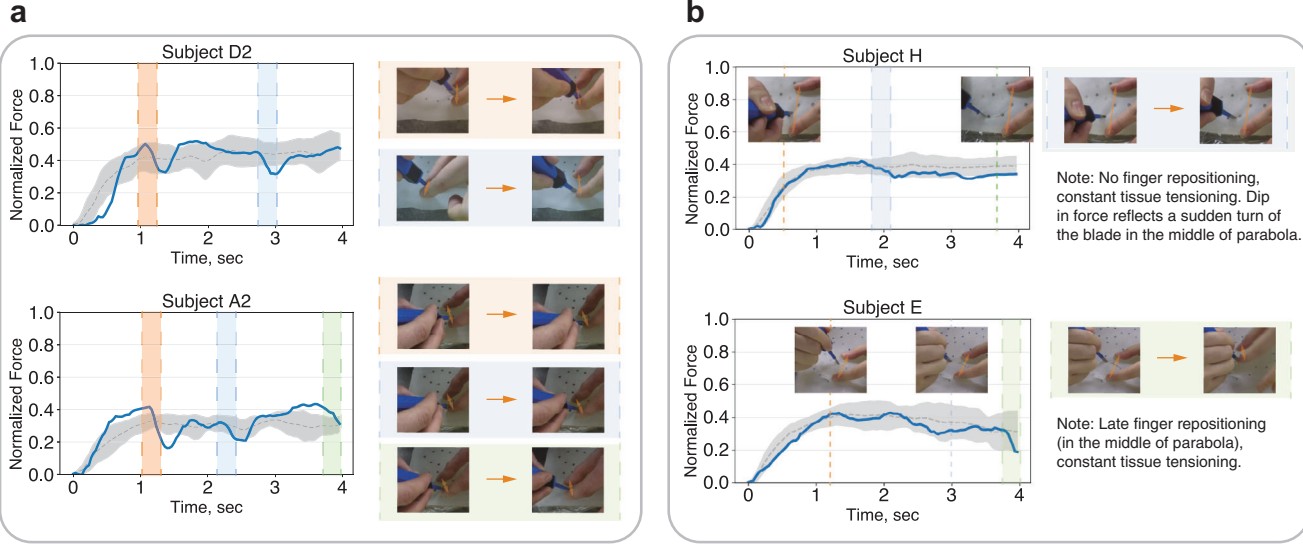

**Fig. 3 Comparison of high scorers from expert evaluations. a** High scorers from expert A evaluation (second trials of subjects *D* and *A*) performed incisions with frequent tissue re-tensioning. **b** High scorers from expert B evaluation (subjects *H* and *E*) executed incisions with constant tissue tensioning. The blue lines are the individual force profiles (depicted on the images), and the grey dotted lines and shaded regions are the mean and standard deviations of force profiles from high scorers' trials.

(*p*-value < 0.05). Therefore, the omnibus Welch's ANOVA (analysis of variance) and Games-Howell post-hoc tests with a family-wise error rate of 0.05 were used. Subjective evaluation by expert A showed no monotonic relationship with any of the described above force-based metrics (Supplementary Fig. 3, blue lines). On the other hand, expert B ratings showed positive monotonic relationship with the mean force, peak force, scaled force and force integral metrics (Supplementary Fig. 3, orange lines). Expert C ratings showed a positive monotonic relationship with the mean force, scaled force and force integral metrics (Supplementary Fig. 3, green lines). In the peak force metric, the middle rated groups (with scores 1 and 2) by expert B showed no significant difference. For expert C, no significant difference is registered between groups with scores 1, 2 and 3 in the mean force and force integral metrics, and groups 2 and 3 in the scaled force metric. In addition, ratings from experts B and C show negative monotonic relationship with the time derivative of force (groups rated with scores 0, 1 by expert B, as well as groups rated with scores 2 and 3 by expert C show no significant difference). Expert D shows a positive monotonic relationship with scaled force, with groups scored 1 and 2 showing no significant difference. No monotonic relationship between the subjective assessment of experts and force variability is registered.

The analysis above suggests that experts B and C reward the incisions that are executed with smooth (i.e. uninterrupted) force profiles of larger amplitude and low overshoot. This observation is in agreement with an intuitive interpretation of the force-based metrics - higher force integral (larger incision forces with longer duration) along with lower force derivative corresponds to "confident" incisions with consistent application of forces throughout the task execution. In the case of expert D, there is an indication that the expert penalizes the excisions with an overshoot in the force profile.

However, the above analysis fails at explaining the implicit criteria of expert A. Figure 3 compares the high scorers from the experts' evaluations. The top scorers from expert A evaluation executed the incisions with distinct frequency of tissue re-tensioning. In contrast, the top scorers from evaluation by experts B and C show noticeable passivity of the non-dominant hand - the surrounded tissues held in constant tension with occasional

finger re-positioning in the later stages of task execution. The inspection of the commentary from expert A (Supplementary Table 2) further suggests that active re-positioning of fingers (or tissue re-tensioning) might be one of the dominant performance criteria employed by the expert. Nevertheless, the traditional force-based metrics fail to identify this rating dimension. In the following section, we show how this problem can be addressed by exploring the parameter space of our probabilistic generative model.

**Elliptical excision force model parameters and behaviour analysis.** The proposed elliptical excision force model (see Methods section for details) encodes the observed cutting behaviour using the following set of parameters with meaningful and intuitive interpretation:

- $v_L$ and $v_U$, which determine the lower and upper excision force levels and characterise the overall amplitude and the spread of the force profile distribution.
- $\sigma_L^2$ and $\sigma_U^2$, which capture the uncertainty of the upper and lower excision force levels and reflect sample-to-sample variability within the force profile.
- transition probability matrix $\mathbf{Q} = \begin{bmatrix} q_{11} & q_{12} \\ q_{21} & q_{22} \end{bmatrix}$, which determines the temporal characteristics of the incision force profile, i.e. the modulation of forces observed in the experiment. Here, $q_{12}$ is the probability of switching from the lower to the upper force level, $q_{21}$ is the probability of switching from the upper to the lower force level, $q_{11}$ and $q_{22}$ are probabilities of remaining in the lower and upper force levels, respectively.

Figure 4 illustrates the effect of the above parameters on the learned behaviour for subjects *H* and *D*. Note that actual incision forces exerted by the subjects have similar mean amplitude (approx. 0.4), but differ in the force envelope - subject *H* shows a tighter distribution in force profiles compared to subject *D*, which is reflected in the corresponding $v_L$ and $v_U$ parameters. In addition, the subjects differ in temporal characteristics of the

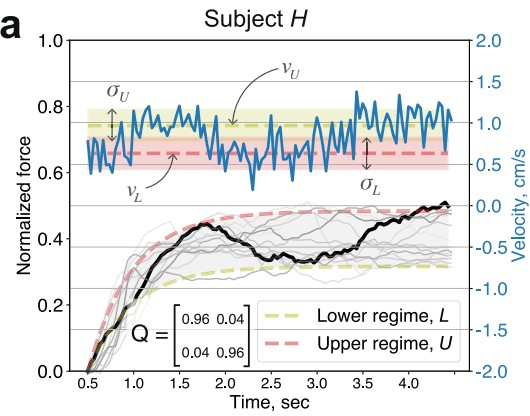
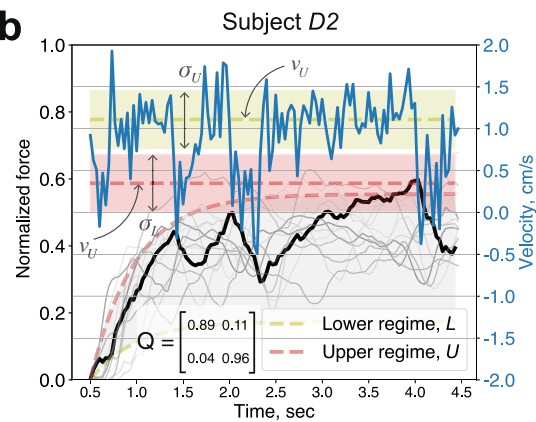

**Fig. 4 Excision forces and model parameters.** Learned model parameters ($v_L, v_U, \sigma_L^2, \sigma_U^2$ and **Q**) from (**a**) subject $H$ and (**b**) second trial of subject $D$, respectively. Thick black lines are synthetic force profiles (generated by the trained model), blue lines are corresponding generated velocity profiles, and semi-transparent grey lines are the actual force profiles used in model training. Note: The pink and green shading corresponds to the standard deviation of velocity at the lower and the upper regimes ($\sigma_L$ and $\sigma_U$), respectively. The grey shading denotes the envelope of the excision forces, defined by the lower and the upper regimes ($v_L$ and $v_U$).

excision forces - subject $H$ shows slow varying modulation between the upper and the lower force levels, whereas subject $D$ shows occasional losses in the excision forces followed by rapid recovery to the upper force level. These characteristics are captured by the transition probability matrix **Q** (Fig. 4).

Figure 5a shows the scatter plot of $v_L$ versus $v_U$ parameters across the subjects (including the surgeons $SA$ and $SB$), and the corresponding distributions of forces for each cluster along the amplitude axis (higher the $v_L$ and $v_U$ parameter values correspond to the higher mean forces). Note that subjects $H$ and $D$ are well aligned along the amplitude axis, as expected. In addition, it should be noted that the axis orthogonal to the amplitude axis describes the width of force envelope, e.g. simultaneous increase in $v_L$ and reduction in $v_U$ corresponds to narrower force profiles, and vice versa (see the effect of $v_L$ and $v_U$ parameters on the force envelope in Fig. 4). As expected, subjects $H$ and $J$, as well as surgeons $SA$ and $SB$ are located in the bottom right corner of Fig. 5a plot, reflecting highly consistent force application with a narrow envelope (Fig. 2).

The proposed model implicitly encodes the descriptive statistics of the excision forces and provides a compact representation of a range of heuristic metrics previously considered in the literature, such as mean forces or force variability. However, our model extends the analysis by explicitly capturing the temporal structure of the behaviour, which is typically lost when descriptive statistics are computed directly. For instance, a close inspection of incision force profiles from subjects $J$ and $H$ (Supplementary Fig. 4a) reveals that subject $J$ executes the incision with a lower amount of modulation of the force amplitude. However, the standard deviation of normalized force profiles (the width of the force envelope) for $J$ and $H$ subjects is identical, $0.056 \pm 0.031$ vs $0.056 \pm 0.036$ ($N = 12$, excision profiles), respectively. In addition, the force profiles from subject $J$ trials exhibit a higher force derivative metric score[11] ($3.7 \pm 0.37$ vs $3.2 \pm 0.53$, $N = 1440$ force samples), which might lead to an incorrect conclusion. Our model correctly captures this temporal characteristic with the transition probability matrix **Q**: the smooth and slowly varying force profile modulation shown by subject $H$ is reflected in the equal and low transition probabilities $q_{21} = q_{12} = 0.037$. In contrast, the imbalance in the transition probabilities for subject $J$ ($q_{12} = 0.124$ and $q_{21} = 0.028$) yields a considerably higher long-term probability of application of a steady excision force ($\pi_U = 0.819$) compared to subject $H$ ($\pi_U = 0.496$). Supplementary Fig. 5 illustrates the

combined effect of transition probabilities and amplitude parameters on the learned excision characteristics for subjects $J$, $H$, $A2$ and $C1$.

The Principal Component Analysis (PCA) of model parameters allows the extraction of meaningful features that characterise the performance. Figure 5b shows the PCA projection of model parameters for each subject on the 2D plot, with highlighted groups along the diagonal axis. The principal component PC1 reflects a simultaneous reduction in the lower force level $v_L$ (Supplementary Fig. 6a) and an increase in the probability of a sudden drop of applied forces $q_{21}$ (Supplementary Fig. 6b). In other words, the higher end of the PC1 axis corresponds to a more frequent and drastic loss of applied force throughout the task execution. The PC2 component reflects the increase in the probability of a sharp rise of excision forces (Supplementary Fig. 6c), i.e. the higher end of the PC2 axis corresponds to a more aggressive brush stroke-like application of excision forces. We call the diagonal axis on PC1 vs PC2 plot an Abruptness feature, as it reflects a degree of discontinuity of the task execution.

The third principal component PC3 corresponds to a reduction of the upper force level $v_U$ (Supplementary Fig. 6d). Note that model parameters whose projection lies on the high ends of PC1 and PC3 would correspond to low overall excision forces (due to low values for $v_L$ and $v_U$ parameters) with frequent switching to a lower force level (due to high probability $q_{21}$). Conversely, the model parameters that are projected to the lower regions of the PC1 and PC3 axes would correspond to high excision forces with rare loss of the applied forces. We call this diagonal axis of the PC1 vs PC3 plot an Energy feature (the higher excision forces applied for a longer duration, the greater the energy of task execution). Figure 5c shows the PC1 vs PC3 plot and groups of subjects aligned along the Energy axis.

Finally, the model parameters that are simultaneously projected on the lower end of the PC1 and on the higher end of the PC3, correspond to highly uniform (due to low probability of $q_{21}$) and highly consistent excision forces with narrow envelope (due to high values of $v_L$ and low values of $v_U$ parameters). We call this diagonal of PC1 vs PC3 plot a Confidence feature. Note that the Confidence axis is orthogonal to the Energy feature, i.e. equally confident excisions can be executed at different energy levels (e.g. subject $J$ and surgeon $SB$), and vice versa (e.g. subject $I$ and surgeon $SA$). Supplementary Fig. 7 shows the plot of the above features against the expert scores.

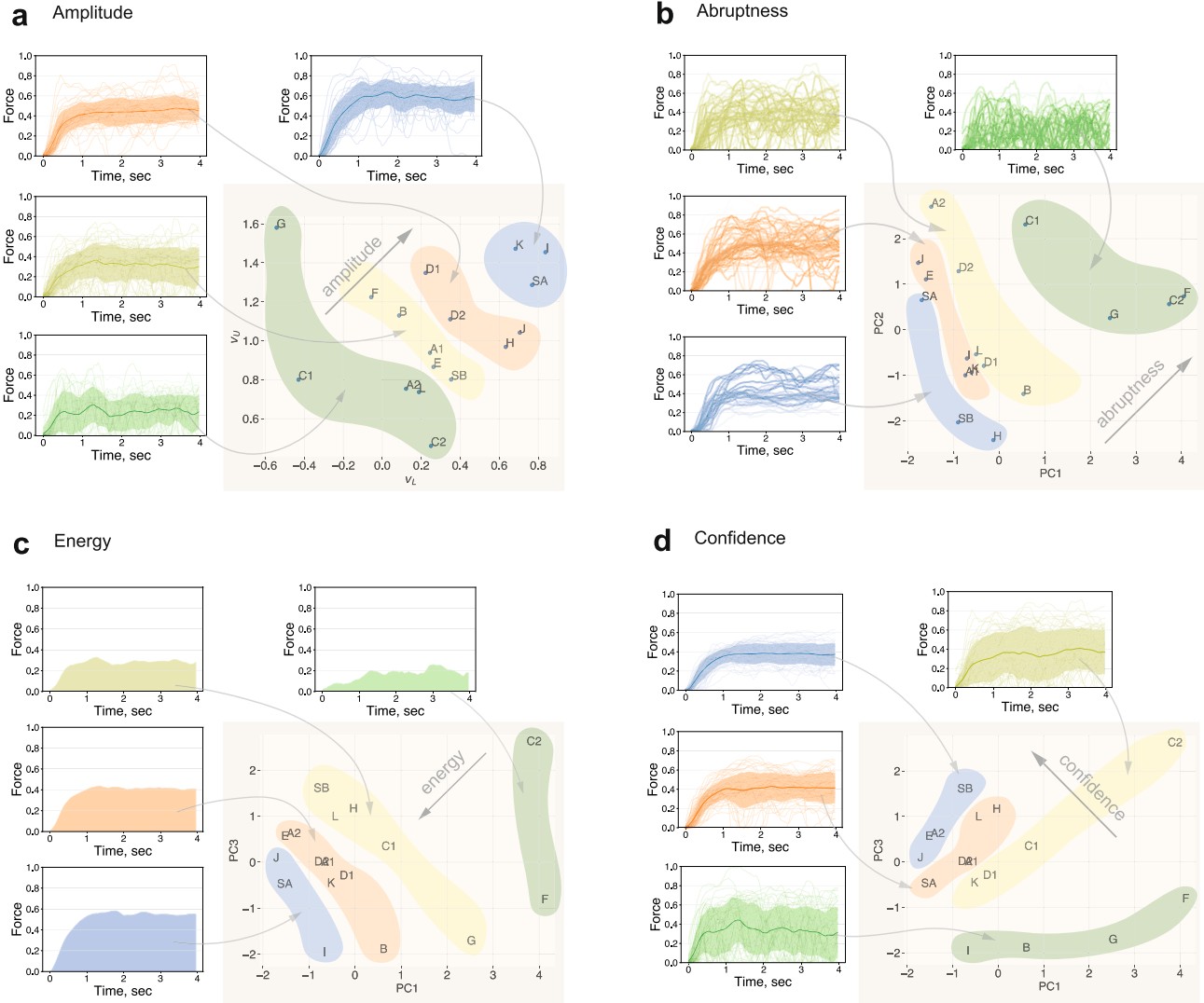

**Fig. 5 Performance analysis using model parameters. a** Parameters $v_L$ and $v_U$ encode the amplitude information of the excision forces (the mean, the standard deviation and the individual force profiles for each highlighted group are shown as solid lines, shaded region and semi-transparent lines, respectively). **b**, **d** The Principal Component Analysis representation of parameter space encodes meaningful features that can characterise the task execution. **b** The diagonal axis on the Principal Component 1 (PC1) vs Principal Component 2 (PC2) plot captures the excision abruptness, characterised by increased probabilities of sudden rises and falls in the applied forces. **c** The PC1 vs Principal Component 3 (PC3) plot captures the Energy feature, characterised by the amplitude and steadiness of the excision forces (the integrals of the mean force profiles for each of the groups is shown here). **d** Orthogonal to the Energy axis is the Confidence feature reflecting the consistent and steady force application (the mean, the standard deviation and the individual force profiles of the highlighted groups are shown as solid lines, shaded region and semi-transparent lines, respectively). Note: Letters A to L correspond to medical students (where numeral indicates the trial), "SA" and "SB" correspond to surgeon A and B, respectively.

**Beyond traditional performance analysis**. We performed correlation analysis to identify whether the inferred parameters of the proposed elliptical excision force model reflect the evaluation score provided by each of the experts. The expert B evaluation scores showed significant ($p$-value < 0.05) Spearman rank-order correlation with $v_L$, $\sigma_L^2$, $q_{11}$ and $q_{22}$ model parameters. The performance evaluation by experts C and D showed significant Spearman's rank correlation with parameters $\sigma_U^2$ and $q_{22}$. In the above analysis, the critical value of 0.446 was used for $N = 15$ observations[26].

Figure 6 (top row) shows the scatter plot of $v_L$ and $v_U$ parameters with a contour plot of linearly interpolated evaluation score provided by each of the experts. The plot suggests that evaluation by expert A is approximately invariant to the overall amplitude of the force profiles, however, it is well aligned with an axis that defines the width of the force envelope. In addition, it

can be seen that the top scorers from the evaluation of expert A cut with higher force envelope width compared to the top scorers from other experts. Note that the top scorers by expert B cut with higher mean force (i.e. subjects are located higher along the $v_L$ and $v_U$ axes) compared to other experts.

Figure 6 (middle row) shows the PCA projection of model parameters across the expert evaluations with highlighted Abruptness feature. It can be seen from the plot, that interpolated evaluations of experts A, B and C are well aligned with the Abruptness axis. Note that the top scorers evaluated by expert A are located further along the axis compared to evaluations from experts B and C, which indicates that expert A rewards task executions with highly pronounced modulation of the excision forces. In contrast, the top scorers from experts B and C are located on the lowest side of the Abruptness feature, suggesting that the experts penalise discontinuous application of excision

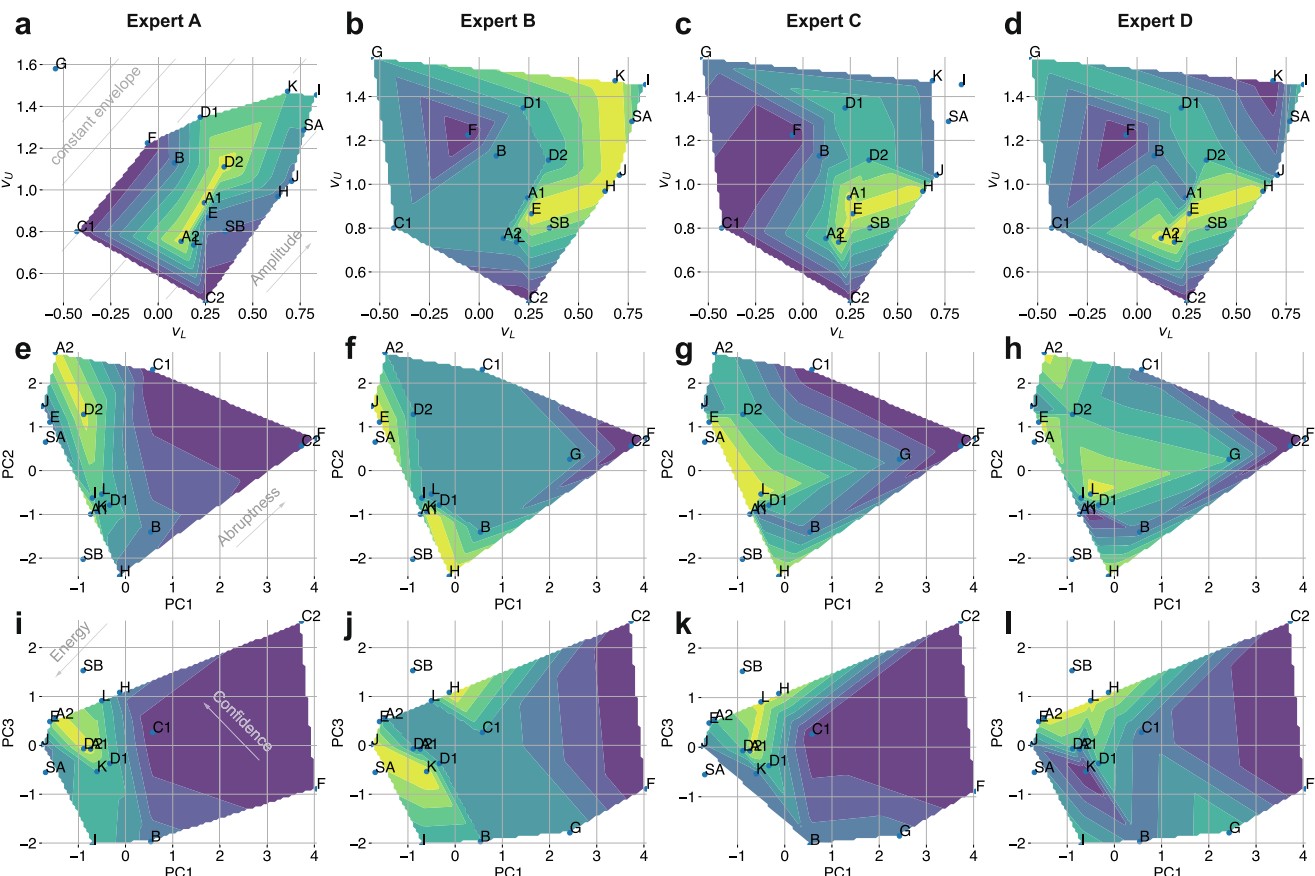

**Fig. 6 Analysis of experts' evaluation criteria with model parameters. a–d** Scatter plot of parameters $v_L$ and $v_U$ with linearly interpolated expert evaluation score (with the brighter region corresponding to the higher score). **e–h** and **i–l** PCA plots of model parameters with linearly interpolated expert evaluation score. Note the subjects $G$ and $I$ were excluded as outliers from the analysis of experts A and C. Note: Letters $A$ to $L$ correspond to medical students (where numeral indicates the trial), "$SA$" and "$SB$" correspond to surgeon A and B, respectively.

forces. Finally, expert D showed no distinctive alignment with the defined axis, which suggests that the Abruptness feature does not reflect the expert's evaluation criteria.

Figure 6 (bottom row) shows the PC1 vs PC3 plot, with corresponding Energy and Confidence features. It can be seen from the PCA plots that expert A rewarded the performers that scored highly along the axis of the Energy feature, as well as moderately along the Confidence axis. This agrees with previous conclusions that expert A values a certain degree of force modulation. In addition, it can be noted that expert B rewarded the performers that executed the task with high energy and high confidence (with an exception of subject $K$). Finally, the evaluation of expert D appears invariant to the Energy axis, however, it is well aligned with the Confidence axis (the top scorers are clustered in the region of the highest confidence score).

The above analysis suggests that, in contrast to other experts, expert A rewards the incisions that are executed with a wider force envelope and an increased amount of switching between the distinct force levels. In the elliptical excision task, such behaviour corresponds to an explicit force modulation due to well-pronounced tissue re-tensioning or finger re-positioning events (Fig. 3). This conclusion agrees with both the additional commentary from the expert (Supplementary Table 2), as well as with the qualitative assessment of force profiles from the distribution of high scorers (Supplementary Fig. 4a).

In summary, the analysis indicates that expert B rewards confident incisions executed with higher energy. Expert C rewards excisions with consistent force application (i.e. narrow

envelope of the force profiles). Both experts B and C penalise interrupted incisions. Finally, according to the analysis, expert D rewards the Confidence feature, but is invariant to the Energy feature, which suggests that overall force amplitude is not part of the expert's evaluation criteria. Importantly, the above analysis is in agreement with conclusions derived from the traditional force-based metrics, yet it offers an additional insight by introducing temporal features into the analysis.

## Discussion

The contributions of this work are threefold. Firstly, we have developed a low-cost easy-to-replicate cutting instrument with an integrated force sensor. Secondly, our experiments using this instrument revealed that the time series of incision forces consists of subject-specific signatures that can reflect the subjective expert evaluation, and can be used for downstream performance analysis and objective surgeon comparisons. Thirdly, we compare the traditional force-based analysis techniques with the proposed superior method of analyzing incision forces.

The collected dataset of elliptical incisions shows a distinct pattern of a step-like response in the cutting force, with noticeable amplitude modulation in the steady-state phase. We found that incision force profiles encode the characteristics relevant to the perceived quality of task execution, and therefore can map the subjective criteria of an expert. The proposed model extends traditional descriptive statistics through a rigorous treatment of the temporal dependency of force measurements and conveniently decomposes the cutting behaviour into amplitude and temporal components. Analysis showed that this decomposition

offers greater flexibility and brings deeper insight into the complex behaviours of surgeons, which are characterized by strong temporal structure.

We intentionally limited the scope of this study to the analysis of incision forces alone. We acknowledge the importance of motion analysis, and regard the role of force measurements as complementary. Nevertheless, it is critical to highlight the practical implications of force-based skill quantification. As accurate motion capture remains prohibitively expensive and difficult to deploy in realistic settings[27,28], the tools and analysis approach described in this work offer an opportunity to explore the composition of surgical skills at a considerably larger scale.

This paper opens up a number of opportunities for future work. Firstly, a comprehensive analysis of the utility of objective performance characterisation using a greater number of participants would be valuable, alongside work investigating skill requirements for different tasks and procedures. Future studies would also benefit from a comprehensive analysis of the learning curve, with a series of repeated trials across the entire cohort. The mapping between these objective measurements and downstream patient outcomes would also be particularly interesting. Moreover, an analysis of the variations in criteria underpinning subjective evaluations of surgeons would be valuable, and it would be interesting to determine if there are specialisation-specific nuances or preferences present using the techniques introduced here. Finally, with minor modifications to sensing hardware, the described method can be applied to studying other complex manipulation skills, such as tissue characterization through palpation, or gentle grasping, where the force modality and its temporal components are also likely to play a dominant role. Finally, the proposed model is particularly promising for the analysis of highly procedural surgical tasks with multiple distinct execution phases, such as suturing. Although we found that two regimes are sufficient for modeling the force measurements in the elliptical excision task, the number of states can be increased for modeling more complex data. Being a hybrid system, our model enables modeling complex nonlinear behaviours with multiple linear dynamical systems. In practice, however, the inference of large number of parameters for switching linear dynamical system can be challenging given limited and noisy measurements.

## Methods

**Experiment**. Twelve right-handed medical students (four female and eight male) and two professional surgeons (both male) were recruited for this study. We labelled medical students with letters A to L, and surgeons with"SA" and "SB" labels (referring to surgeon A and B, respectively). Only three subjects (A, C and D) repeated the trials (two months after the first trial). Subjects that repeated the trials have a numeral in the label indicating the trial order (e.g. "A2" means the second trial of subject A). None of the student participants had any prior experience in surgical cutting tasks. The study was approved by the University of Edinburgh, School of Informatics, Informatics Ethics panel. All participants provided written informed consent to participate in this study.

The participants were asked to perform a series of 6 elliptical excisions on the phantom using the sensorised cutting tool (Fig. 1d). Before each trial, a new blade (Swann-Morton No. 10) was mounted to the cutting tool. After receiving the task instructions, participants were familiarized with the experimental setup, cutting tool ergonomics, phantom mechanical properties, etc. Next, each subject was asked to rehearse the described task using a dedicated sacrificial phantom. During the trials, the cutting forces that act on the blade in the direction of cutting were recorded at a fixed frequency of 30 Hz. Finally, at the end of the trials, each participant was asked to complete a post-study questionnaire.

**Data measurement**. Each participant performed six elliptical excisions as a part of the task, yielding 12 force profiles per trial (each excision consists of upper and lower cuts). The recorded profiles were time-aligned and cropped to a fixed duration of 120 samples or 4 seconds (at a sampling rate of 30 Hz). Finally, the samples were normalized to the maximum force value in the entire dataset.

Given the normalized force profiles $f(t)$ (Supplementary Fig. 4a), the virtual displacement profiles $x(t)$ (Supplementary Fig. 4b) were obtained by solving the

differential equation for the Maxwell model, equation (1), as follows:

$$x(t) = \frac{f(t)}{E} + \frac{1}{\eta} \int_0^T f(t)\, dt \tag{1}$$

where $f(t)$ is the corresponding force profile, $T$ is the duration of the force profile, $\eta = 0.5$ N s cm$^{-1}$ and $E = 1$ N cm$^{-1}$ are Maxwell model's damping and spring coefficients, respectively.

The corresponding virtual velocity profiles $\dot{x}(t)$ (Supplementary Fig. 4c) were obtained by approximating the time derivative of $x(t)$ using the finite difference method with a step size $dt = 0.033$.

**Elliptical excision force model**. The collected incision force profiles show temporal features that can characterize the cutting behaviour. For example, the characteristic dip in an incision force profile (Fig. 1b) might reflect a dynamic change in the configuration of the blade, tissue tensioning applied by the non-dominant hand, or both. Here, we propose a generative model that captures these subject-specific temporal features in the force profiles and enables the disentanglement of skill from incision force analysis.

Figure 1 a shows the approximate model of the task of cutting a viscoelastic phantom as a continuous blade's movement through a Maxwell body. In the context of this approximation, the Maxwell model[29] relates the actual incision force $f(t)$ to a "virtual" velocity of the blade $\dot{x}(t)$, as follows:

$$\eta \dot{x}(t) = f(t) + \frac{\eta}{E} \dot{f}(t) \tag{2}$$

where $\dot{f}(t)$ is the time derivative of the force, and $\eta$ and $E$ are the Maxwell model's damping and spring coefficients, respectively.

By taking the Laplace transform of equation (2) and rearranging the terms, we obtain the transfer function $G(s)$, which relates a virtual blade's displacement $X(s)$ and the actual force $F(s)$, as follows:

$$G(s) = \frac{F(s)}{X(s)} = \frac{\eta s}{\frac{\eta}{E} s + 1} \tag{3}$$

The above transfer function indicates that the model exhibits high-pass characteristics in the force response to the displacement input. This predicts an exponential decay of force with a time constant $\frac{\eta}{E}$, as response to a unit step displacement. Importantly, this also predicts a step-like response in the force to a ramp-like displacement input, and therefore, the observed cutting force profiles can be described as a response to a continuous virtual scalpel displacement $x(t)$ at a constant velocity. As such, this model represents an elliptical excision process as a virtual hybrid system with $K$ linear regimes, in which the blade velocity $\dot{x}(t) = v_k$ is feedback-regulated by means of switching between the discrete regimes $v_1,...,v_K$. In this work, we show that such formulation can bring a greater insight into the analysis of surgical skill when compared to the descriptive statistics approach more commonly applied in this area. In the next section, we focus on the problem of inferring the parameters of our model from force measurements.

**Excision as a switching linear dynamical system**. The switching linear dynamical system[30–35] is an example of a broader class of hybrid system, in which globally nonlinear dynamics are approximated by a series of linear systems. In the generative model of a switching linear dynamical system, the switching between each of its $K$ linear regimes is described by a discrete hidden state variable $s_t \in \{1,...,K\}$. The evolution of $s_t$ is characterized by a $K \times K$ transition matrix $\mathbf{Q}$ that captures the probabilities of state transitions, i.e. $P(s_t|s_{t-1})$. The continuous hidden state vector $\mathbf{z}_t \in \mathbb{R}^D$ evolves according to a $D \times D$ dynamics matrix $\mathbf{A}$, and the observation vector $\mathbf{y}_t \in \mathbb{R}^L$ is generated according to an $L \times D$ observation matrix $\mathbf{C}$, as follows:

$$\mathbf{z}_t = \mathbf{A}^{(k)}\mathbf{z}_{t-1} + \mathbf{w}_t^{(k)}, \tag{4}$$

$$\mathbf{y}_t = \mathbf{C}^{(k)}\mathbf{z}_t + \mathbf{v}_t^{(k)}. \tag{5}$$

where $\mathbf{A}^{(k)}$ and $\mathbf{C}^{(k)}$ are associated with a regime $s_t = k$, and $\mathbf{w}_t^{(k)}$ and $\mathbf{v}_t^{(k)}$ are the disturbance and observation noise, respectively.

In this work, we model the elliptical excision process with two discrete linear regimes, $k \in \{L,U\}$. Each regime corresponds to a constant virtual velocity of the blade, and satisfies $v_L < v_U$ (we call $L$ — a lower regime, and $U$ — an upper regime). For each of these linear regimes, we model the uncertainty in the constant velocity as $\tilde{v}_k \sim \mathcal{N}(v_k, \sigma_k^2)$, where $\sigma_k^2$ is the variance of the velocity noise in the regime $k$.

The continuous hidden state vector $\mathbf{z}_t = \begin{bmatrix} g_t \\ x_t \\ 1 \end{bmatrix}$, comprises $g_t$ and $x_t$, the latent cutting force and virtual displacement of the blade at time step $t$, respectively. Since we only measure the cutting force, the observable $y_t$ is a scalar that represents the force measurement at time step $t$. The continuous dynamics in the linear regime $k$ is $\mathbf{A}^{(k)} = \begin{bmatrix} \alpha & \beta & 0 \\ 0 & 0 & \tilde{v}_k \\ 0 & 0 & 0 \end{bmatrix}$, where constants $\alpha$ and $\beta$ define the displacement-to-force relationship of the Maxwell model, and are found by transforming the transfer

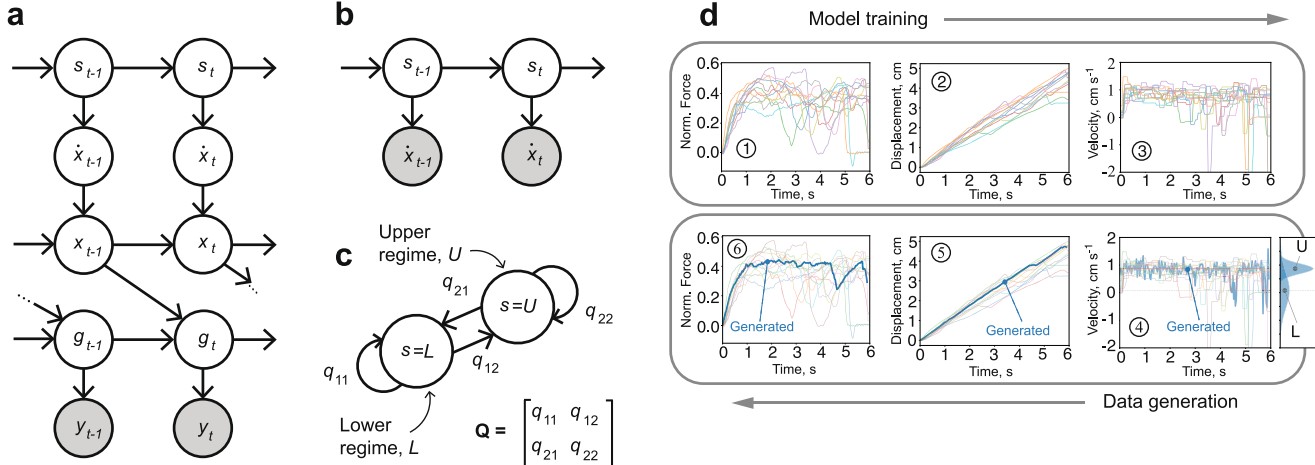

**Fig. 7 The elliptical excision force model. a** A graphical model representation of the generative model, where $s$ is a discrete state, $\dot{x}$ is blade's velocity, $x$ is blade's displacement, $g$ is excision force, $y$ is force measurement and $t$ is a time step. Shaded nodes represent the observed variables. **b** Hidden Markov Model (HMM) with a hidden discrete state $s_t$ (the cutting regime at time step $t$), and an observable virtual velocity $\dot{x}_t$. **c** Markov chain with two cutting regimes defined by the transition matrix **Q**. **d** Model fitting (1, 2 and 3) and data generation processes (4, 5 and 6). (1) Actual measurements of forces collected during the trials. (2) The virtual displacement derived from the force measurements using the Maxwell model. (3) The virtual velocity profiles (finite differences of the displacement profiles) are used to train the HMM. (4) The velocity sampled from the trained HMM (blue line). (5) and (6) The synthetic displacement and force (blue lines), generated by the model.

function, equation (3), into the equivalent state space form. The observation matrix in the linear regime $k$ is $\mathbf{C}^{(k)} = \begin{bmatrix} \gamma & \delta & 0 \end{bmatrix}$, where $\gamma$ and $\delta$ are the observation constants from the state space representation of the Maxwell model's transfer function. In this work, we set the spring constant $E = 1\,\mathrm{N\,cm^{-1}}$ and the damping coefficient $\eta = 0.5\,\mathrm{N\,s\,cm^{-1}}$, which yield $\alpha = -2$, $\beta = 1$, $\gamma = -2$ and $\delta = 1$ constant values. The parameters were chosen such that estimated displacements approximately match the actual distance travelled by the scalpel. Finally, given the uncertainty captured in the velocity $\tilde{v}_k$, we can further assume the disturbance-free dynamics ($\mathbf{w}_t^{(k)}$ is zero vector) and noise-free observations ($\mathbf{v}_t^{(k)} = 0$).

A graphical representation of this generative model is shown in Fig. 7a. There are several ways to infer the parameters of this class of models from observations. For example, the variational approach to learning in switching linear dynamical systems[34] approximates the posterior probabilities of the hidden states by optimizing evidence lower bound. In this study, we bypass the inference of discrete hidden states $s_t$ by assuming that velocities $\dot{x}(t)$ are fully observable under the assumption of the Maxwell model (Fig. 7b). This turns the switching linear dynamical system inference into a problem of learning an HMM[36], fully characterized by transition probability matrix **Q** (Fig. 7c) and the emission probabilities defined by $v_k$ and $\sigma_k^2$, for each of the linear regimes $k$. Given the virtual velocity profiles $\dot{x}(t)$, this model can be easily fit using the Expectation-Maximization algorithm[37].

Figure 7d provides an overview of the model fitting process. First, the virtual displacement profiles are derived from the force measurements using the inverse of the transfer function, specified by Maxwell model parameters, equation (3). Then, the obtained displacement profiles are numerically differentiated for estimation of the virtual velocities $\dot{x}(t)$. Finally, the obtained virtual velocity profiles are used to fit an HMM with the Expectation-Maximization algorithm. (Examples of incision forces generated by the model when fit to each of the medical students are shown in Supplementary Fig. 8).

**Sensorized cutting instrument**. We constructed a uniaxial force sensor based on Texas Instrument's LDC1612 inductance-to-digital converter (LDC) and a 3D printed flexible element. The LDC provides reliable position measurements at submicron resolution[38], which in combination with a flexible element with a known stress-strain characteristic, enables the construction of displacement-based force sensors. The LDC measures the distance between a conductive target and an inductive coil using the resonant sensing principle. The inductive coil in parallel with the capacitor forms a resonant circuit in which the alternating current flowing through the inductor generates an alternating magnetic field. As a result of Faraday's law, the alternating magnetic field induces eddy currents on the surface of the conductive target as a function of the target displacement. As per Lenz's law, these eddy currents create an opposing magnetic field that reduces the nominal inductance of the resonant circuit, and hence, increases the resonant frequency. The LDC measures this frequency shift and thus provides information about the target's displacement with respect to the inductor. By fixing the target to the free end of the flexure with a known stress-strain characteristic, a displacement measurement can be transformed into a force measurement.

The designed cutting tool consists of two key components, 1) a printed circuit board with an inductive coil, and 2) a flexure with a conductive target. The schematic for the uniaxial force sensor is shown in Supplementary Fig. 9b. The inductor is implemented as a circular planar coil of 8 mm diameter as shown in Supplementary Fig. 9a. In the rest configuration of the flexure, the effective 8.6 μH inductor (in parallel with 330 pF capacitor) focuses the alternating magnetic of 2.985 MHz frequency into the conductive target located 3.4 mm below. In our design, we used 10 mm square aluminium film of 0.2 mm thickness. The displacement range of the target is restricted to 1.6 mm, with a minimum distance to the inductor of 1.8 mm. When the flexure is at its maximum displacement configuration, the resonant frequency shifts from 2.985 MHz to 3.025 MHz (40 kHz shift, 1.3% of the nominal resonance at zero displacement). According to ref. [39], the maximum effective resolution achievable with the given frequency variation is 14-15 bits. The dimensions of the printed circuit board are 100 mm x 13.5 mm. The 4-layer board incorporates differential sensor coils, the LDC1612 inductance-to-digital converter, an MSP430F5528 microcontroller, power supply circuitry and a USB connector. The microcontroller configures the LDC via the I2C interface, implements USB Communication Device Class, processes and streams sensor data to a host computer.

The displacement is established by a one-piece 3D printed flexure, in which the free end displaces the conductive target under the presence of external force. As with any displacement-based force sensor, one of the main challenges is to maximize the stiffness of the flexure, while achieving the desired sensitivity. 3D printing provides a relatively easy way of experimenting with various design parameters, such as stiffness, strength, and geometry, as well as printing process parameters, such as material, printing orientation, etc. In this study, we use a blade flexure with design parameters shown in Supplementary Fig. 9b. The flexure was 3D printed with an Ultimaker 3 Extended printer using PLA thermoplastic, 0.2 mm layer height, 20% infill (triangle pattern) and 0.4 mm nozzle diameter. The extruder temperature was set to 205 °C, the travel speed was set to 70 mm per second and the perimeter layers were set to 3. The printing was done at room temperature controlled in a range between 19 and 21 °C. With these settings, the printed element was approximately 50 microns wider in XY direction.

Supplementary Fig. 9c shows the results of the incremental load test. During the test, a fully assembled device was incrementally loaded by ten 100 g calibrated weights (i.e. from 0.98 N to 9.8 N). The load was applied at the midpoint of the blade interface. The hysteresis (defined as the maximum difference between loading and unloading samples relative to the full-scale output) is 3.9%. The dotted line on the graph represents the linear least squares fit to the loading curve. The maximum deviation from the linear fit (non-linearity) is 1.4% of the full-scale output and the sensitivity of the sensor is 3752 counts per newton. Finally, the measured accuracy (maximum standard deviation of sensor output at the maximum measured load and relative to the maximum measured load, i.e. to 9.8 N) is 0.58%.

**Tissue phantom**. Supplementary Fig. 9d illustrates the design and material composition of the multilayered phantom used in this study. The design consists of a gelatin base that simulates the recoil of subcutaneous tissues, and a stack of three silicone layers that mimic the mechanical properties of human skin. The outer silicon layer is reinforced by pre-tensioned power mesh fabric that increases the

tear strength of a sample. The gelatin base and silicon layers are coupled through a thin layer of an ultrasound gel. The fully assembled phantom has dimensions of 160 mm x 160 mm x 30 mm.

The fabrication of each phantom comprised of the following procedure. 64 g of gelatin powder (240 Bloom) was spread across 640 ml of cold water and left unstirred for 20 min, then simmered and stirred until fully dissolved. The liquid was poured into a 3D printed mould (160 mm x 160 mm x 25 mm volume container) wrapped in cellophane film and was left to solidify overnight in a refrigerator.

Next, a square piece of power mesh fabric (180 mm x 180 mm) was secured to the working surface under a slight amount of tension. 20 ml of two-part silicone rubber (Smooth-On Ecoflex$^{TM}$ 00-30, shore hardness 30) was thoroughly mixed in a 1:1 ratio for 2 min and poured onto the center of stretched fabric in the series of three pours. The silicone-saturated mesh was then left for 45 min to cure. When cured, the next layer of 20 ml silicone (Smooth-On Ecoflex$^{TM}$ GEL with shore hardness 000-35) was mixed and poured over. Finally, the second batch of 25 ml Smooth-On Ecoflex$^{TM}$ 00-30 was poured over the pre-cured silicone layers. The silicone sample was left to cure for 4 h.

The cured silicone sample was placed on the full set gelatin base with a power mesh-reinforced layer presenting the skin surface. The remaining edges of the power mesh are trimmed to match the surface area of the phantom. The fully assembled phantom is stored in a refrigerator prior to each experiment.

The design of the phantom was selected after extensive validation with a single experienced surgeon, and selected for its realistic viscoelastic properties. A total of seven phantom designs were evaluated according to the perceived realism of pressing, stretching, pinching and cutting the phantom surface. All evaluated designs consisted of a gelatin base with 100 g per litre concentration and varying combinations of silicone layers. We have chosen Smooth-On Ecoflex$^{TM}$ Gel, Smooth-On Ecoflex$^{TM}$ 00-30 and Smooth-On Dragon Skin$^{TM}$ (shore hardness 10A) silicone rubbers to represent very soft, soft and hard phantom layers, respectively. Supplementary Table 5 shows the phantom design ranking (from least to most realistic). A few summary points:

- Softer silicone rubbers (shore hardness < 30) appear more realistic.
- The combination of silicone layers with varying hardness increases realism. Single-layer designs were scored lowest, while three-layer designs were rated as most realistic.
- The hardness gradient (with a harder outer layer) plays a role in the realism of shear loads (e.g. stretching the skin).
- The hardness of the bottom layer plays role in pressing load and can mimic the age of the skin.

**Statistics and reproducibility**. The statistical analysis was performed using open-source Python libraries SciPy (https://scipy.org/) and Pingouin (https://pingouin-stats.org/build/html/index.html). The elliptical excision force model was trained using open-source Python package hhmlearn (https://hmmlearn.readthedocs.io/en/stable/). For reproducibility, all data processing, analysis, modeling and figure generation routines were written using Jupyter Notebook.

## Data availability

CAD files required to replicate the instrument, measurement data from the sensorised instrument and code generating the figures (Jupyter Notebook) are made available publicly via https://github.com/straizys/elliptical-excision-force-model.

## Code availability

The analysis routines are made publicly available via on https://github.com/straizys/elliptical-excision-force-model.

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

## Acknowledgements

We are grateful to Drs. Felicity Mehendale and Aidan Roche who advised this study and provided us with insightful discussion. We thank each participant who volunteered to take part in this study. S.R. acknowledges support in the form of a grant from the UKRI Strategic Priorities Fund to the UKRI Research Node on Trustworthy Autonomous Systems Governance and Regulation (EP/V026607/1,2020-2024).

## Author contributions

A.S. designed and fabricated the sensorised instrument and tissue phantom. A.S. conducted the experiments and performed data analysis and modelling with major contributions from M.B. A.S. wrote the article with contributions from all authors. M.B. and S.R. supervised the project. P.M.B. discussed the methods and results.

## Competing interests

The authors declare no competing interests.
