## [Peer review file · Communications Engineering]

Reviewers' comments:

Reviewer #1 (Remarks to the Author):

This study evaluates the utility of force measures to discriminate subjects with different levels of surgical skill. Using a sensorised scalpel, force measurements along the direction of cutting with the scalpel were captured and modeled as a generative linear dynamical system. 12 medical students were recruited to use the scalpel to make an elliptical incision six times.

1. There are some details missing. The Results section discusses surgeons and subjects. Are they different people not mentioned in the Methods?
2. Within what duration did subjects other than A, C, and D repeat the trials?
3. For subjects A, C, and D - did they repeat all 6 trials after 2 months or was it the original complement of six trials that were spread out over 2 months? If it is the former, then the analysis of data after 2 months should be separated from the analysis of data during the original data collection. ANOVA assumes independence of observations.
4. Figure 3 - what are subplots a to d? Naming the subplots can clarify. Are they ratings for each subject by each of the four expert surgeon raters?
5. The section on 'Limitations of traditional performance metrics' - second and third paragraphs - to which figures or tables is this text referring?
6. Supplementary Figure 2 is confusing. Were data recorded from practising/expert surgeons?
7. Supplementary Table 3 indicates a 6-point rating scale (0 to 5), where as the text mentions a 4-point scale (0 to 4).
8. Figure 5 is overloaded with information, with little in the form of a legend or explanation to help the reader digest it. For example, the lines in plots on the second and third rows are apparently color coded but no legend is provided for it.
9. The analysis is interesting because it extracts features that correspond to variation across raters and explain it. Validating the metrics in a new sample of subjects would be relevant, (it is not needed for this manuscript to be complete). What is the distribution of the metrics described in the paragraphs of text discussing Figure 5 across the subjects / trials with different scores?

Reviewer #2 (Remarks to the Author):

Summary:

- Novel low-cost sensorised scalpel to measure forces at the tool-tissue interface during cutting tasks
- Data recording of an elliptical excision task on a skin-mimicking phantom (12 participants (medical students) and 12 recordings per participant, some of them performed a 2nd trial)
- Derivation of novel force-based metrics to analyze and assess the cutting skill, based on a Hidden Markov Model with two states (cutting with high or low velocity)

Relevance

It addresses a relevant topic in the field of surgical skill analysis by taking force measurements and associated metrics into account and is of interest for Nature Communications Engineering.

Originality & Soundness

The paper presents interesting work. However, it is in a preliminary state and the evaluation as well as the presented results have some flaws (see detailed comments). Language, figures and references are sound.

Clarity of presentation

The paper is clearly written and well structured. It provides a detailed introduction to the basic principles of the approach.

I think it is a nice idea, but I see some substantial points with it that would argue against a CommsEng publication due to several limitations, in its current form it is more suited for a conference proceeding.

Detailed comments (weaknesses):

- Manual assessment of each subject by four raters based only on subjective criteria, why did you not use a formal assessment for surgical skills, e.g. the objective structured assessment of technical skills (OSATS)? Very inconsistent annotation, was there any annotation protocol?
- No formal analysis of the inter-rater agreement (which seems to be very limited)
- One rating per subject instead of one rating per recording: Is it valid to assume that all recordings of one subject demonstrate the same level of skill? Figure 2 does not support this assumption (e.g. first vs 2nd trial)
- Why did some participants perform a 2nd trial and not the whole group?
- Verbose attempt to explain each rater's decisions by means of the derived metrics when it is not clear whether those ratings are meaningful after all
- The recordings of two expert surgeons are not considered during the analysis. How would the proposed metrics describe their skills? Experience level of practicing surgeons not clear. There seems to be also a large deviation between experts according to figure 2.
- Unclear whether the metrics can quantify subtle differences regarding the skill of two subjects.
- Novel metrics are tailored to the elliptical excision task; unclear how similar metrics can be derived for other and maybe more complex tasks
- All in all, no convincing evidence for the validity of the proposed metrics

Thank you for the feedback on our paper. We are grateful to the editors and reviewers for their time and valuable feedback, which we believe has resulted in a much-improved paper. Specifically, we have revised the paper to include:

- Additional methodological information (clarification around experiments, repeated trials, expert analysis, etc.)
- The inclusion of rater reliability analysis
- Clarification regarding the value of the model and derived features when it comes to identifying similar and distinct behaviour preference differences across trainee and expert surgeons.
- An improved motivation and more nuanced discussion around the contribution of this work.

Regarding the latter, the core contributions of this work are to highlight the value and importance of the force modality in the behaviours underpinning surgical skill, the introduction of an open-source tool and method to analyse this information and to show that this analysis is more informative than classical statistics that neglect the temporal aspects and behaviour transitions commonly seen in dexterous manipulation tasks like cutting.

Importantly, we have revised the paper to stress that it is not our intent to identify a new metric that captures the skill level of a surgeon or to replace traditional assessment approaches. The former is a highly subjective goal that is likely to vary depending on the task, the raters and numerous other contextual factors. Instead, our aim is to show that latent factors derived from forces can be mapped to subjective expert ratings, and provide a quantitative means of explaining their decisions and the rationale behind these.

We believe this is an extremely valuable finding and that it opens the door to a number of downstream studies in future work. We respond in detail to reviewer feedback below.

Reviewer 1

Thank you for your detailed review and valuable feedback. We have added a number of details in response to your queries and believe this has greatly improved the clarity of our work.

1. There are some details missing. The Results section discusses surgeons and subjects. Are they different people not mentioned in the Methods?

Response: We refer to each participant (student or surgeon) as a subject. Each subject has a label, namely *A-L* for twelve students, and *SA* and *SB* for surgeon A and surgeon B, respectively. Subjects that repeated the trials have a numeral in the label indicating the trial order (e.g. “A2” means the second trial of subject A). We added clarification to the Methods section (Experiment subsection, paragraph 1).

2. Within what duration did subjects other than A, C, and D repeat the trials?

Response: No subjects other than A, C and D have repeated the trials. We added missing details to the Methods section (Experiment subsection, paragraph 1).

3. For subjects A, C, and D - did they repeat all 6 trials after 2 months or was it the original complement of six trials that were spread out over 2 months? If it is the former, then the analysis of data after 2 months should be separated from the analysis of data during the original data collection. ANOVA assumes independence of observations.

Response: Subjects A, C and D repeated the trial (i.e. all 6 elliptical excisions) after two months. In our analysis, we consider each trial (i.e. the series of 6 elliptical excisions), including the repeated ones, as an independent trial. The experts scored all 15 trials (12 original trials, plus 3 repeated) independently, without any knowledge about which trial was repeated. We added necessary clarification to the “Subjective evaluation of the incision skills” section (paragraph 1).

4. Figure 3 - what are subplots a to d? Naming the subplots can clarify. Are they ratings for each subject by each of the four expert surgeon raters?

Response: Yes, the subplots of Fig. 3 are the ratings for each subject by each of the four experts. We added the missing subtitles to Fig. 3. (We moved this figure to the Supplementary Information document, as Supplementary Fig. 5)

5. The section on 'Limitations of traditional performance metrics' - second and third paragraphs - to which figures or tables is this text referring?

Response: We added Supplementary Fig. 4 to the Supplementary Information document to support the analysis and provided references to the figure in the text (“Limitations of traditional performance analysis” section, paragraphs 1 and 2).

6. Supplementary Figure 2 is confusing. Were data recorded from practising/expert surgeons?

Response: The labels A-L indicate the forces from medical students, and labels “SA” and “SB” correspond to the forces from surgeons A and B, respectively. We added missing clarification to the Supplementary Fig. 2 caption.

7. Supplementary Table 3 indicates a 6-point rating scale (0 to 5), where as the text mentions a 4-point scale (0 to 4).

Response: We thank the reviewer for pointing out this issue! Expert B provided the rating using both a 0-3 scale and an additional 0-5 scale provided of their own accord, and we accidentally included the latter. We have now fixed this issue. (see Supplementary Table 1 and

Supplementary Figure 5). The changes due to this fix are reflected in the sections “Subjective evaluation of the incision skills” (paragraph 2) and “Limitations of traditional performance analysis” (paragraph 2).

8. Figure 5 is overloaded with information, with little in the form of a legend or explanation to help the reader digest it. For example, the lines in plots on the second and third rows are apparently color coded but no legend is provided for it.

Response: We simplified the figure by removing the parallel axis plots - the radar chart conveys the same information to the reader.

9. The analysis is interesting because it extracts features that correspond to variation across raters and explain it. Validating the metrics in a new sample of subjects would be relevant, (it is not needed for this manuscript to be complete). What is the distribution of the metrics described in the paragraphs of text discussing Figure 5 across the subjects / trials with different scores?

Response: We thank the reviewer for recognising the value of our method in mapping features to rater variation. We provide the distribution of metrics across the expert scores in Supplementary Figure 7.

Reviewer 2

1. Manual assessment of each subject by four raters based only on subjective criteria, why did you not use a formal assessment for surgical skills, e.g. the objective structured assessment of technical skills (OSATS)? Very inconsistent annotation, was there any annotation protocol?

Response: Thank you for this excellent question! This raises an important area in need of clarification around our focus, and we have substantially revised our discussion of the intent of this work to address this. As noted by the reviewer, current surgical assessment practice relies strongly on a structured assessment process (e.g. OSATS). This has the benefit of improving agreement and arguably makes an intrinsically subjective rating process more objective, but also does suffer from a number of drawbacks. Specifically, this approach is vulnerable to a “wisdom of the crowds” effect, for dominant personalities to bias the rating decisions along certain dimensions or factors and for underlying factors of importance to be neglected entirely.

Regardless of the underlying assessment strategy used, our work explores the role that forces may play in the perception of surgical skill and shows that this is a sensory modality of value when seeking an objective, quantitative measure to link to potentially subjective assessment practices or even downstream outcomes. We deliberately chose a rating approach that would result in subjective assessment to highlight the fact that our method can use force information to

model the subject-specific characteristics of performance that drive the *perception* of skill across raters. Importantly, our modeling process could be used in conjunction with OSATS or any other rating process. Our goal is not to identify a single metric that encapsulates surgical skill. Rather, we aim to show that forces contain subject-specific information that can align with scoring criteria of the raters. We have added additional detail to clarify this difference of focus (“Subjective evaluation of the incision skills” subsection, paragraph 3).

2. No formal analysis of the inter-rater agreement (which seems to be very limited)

Response: We have added ICC2 (two-way random, single measures) analysis (subsection “Subjective evaluation of the incision skills”, paragraph 1 and Supplementary Fig. 6). However, as noted above, the ability of our approach to disentangle highly subjective and differing opinions using the latent features of force profiles is our core contribution, not the development of a single overarching skill metric.

3. One rating per subject instead of one rating per recording: Is it valid to assume that all recordings of one subject demonstrate the same level of skill? Figure 2 does not support this assumption (e.g. first vs 2nd trial)

Response: In our experiment, the task constituted a continuous execution of 6 excisions, which we assume are performed at a given level of the skill. Next, we assume that repeated trials of the task might be executed with different levels of proficiency, e.g. due to acquired familiarity. Our data (incision forces and videos) confirm that our assumption is valid - there is negligible intra-trial variation but considerable variation across the trials (see Supplementary Figure 1a).

4. Why did some participants perform a 2nd trial and not the whole group?

Response: At the time of the experiment, participant availability was limited by local COVID-19 related restrictions. We added the suggestion to expand the study on repeated trials in future work (Discussion section, paragraph 4).

5. Verbose attempt to explain each rater's decisions by means of the derived metrics when it is not clear whether those ratings are meaningful after all

Response: The proposed generative model encodes the elliptical excision force profiles with a small set of meaningful parameters (see updated “Elliptical excision force model parameters and objective performance analysis” subsection, Fig. 4 and Fig. 5 for more details). These parameters model the distinct characteristics of excision force traces, such as amplitude and variability (that reflect the effort and consistency of force application, respectively), as well as switching characteristics (which reflect the flow of task execution).

One can use our model parameters to construct meaningful and insightful features. For example, consider the “consistency” feature: the difference between v_U and v_L parameters defines the width of the force envelope (see Fig. 4), which reflects the consistency of force

application during excision. In this work, we investigate, whether experts' rating (which is based on the video of task execution) can be aligned with such features, derived from force measurements.

In addition to “engineered” features, we explored the latent representation of the parameter space to identify the dominant features from the data. For example, the identified “abruptness” feature correlates with 1) a model parameter that defines the mean force profile in the lower regime (v_L) and 2) a model parameter that defines the probability of transitioning from a higher to a lower cutting regime (q_{21}). Thus, it reflects the degree of abruptness in task execution, when excision is performed in a series of rapid, short cuts rather than in one continuous cut. As with the “consistency” feature above, the “abruptness” feature has a meaningful and straightforward interpretation.

We expanded the “Elliptical excision force model parameters and objective performance analysis” subsection (included additional Fig. 5, to illustrate the model parameter space and corresponding behaviour types) and updated the “Beyond traditional performance analysis” subsection to clarify the point discussed above.

6. The recordings of two expert surgeons are not considered during the analysis. How would the proposed metrics describe their skills? Experience level of practicing surgeons not clear. There seems to be also a large deviation between experts according to figure 2.

Response: We thank the reviewer for this question. Fig. 6 shows both surgeon A and surgeon B (labelled as “SA” and “SB”, respectively). The noted deviation between experts in Fig. 2 highlights one of the limitations of traditional metrics we aimed to point out. Our analysis reveals that the surgeons didn't deviate much at all: the only considerable differences are their mean force level and the amount of modulation (Fig. 2, note the characteristic dip in forces from surgeon A when compared to a uniform force profile from surgeon B), otherwise they performed very similarly.

Let us first consider the force envelope width, which is notably similar between the surgeons. Indeed, our model captures this similarity - Fig. 6 (row 2) shows that “SA” and “SB” are almost aligned along the “constant envelope width” axis. Also, note that “SA” lies higher along both v_L and v_U parameter axes, which reflects the higher mean forces exhibited by surgeon A when compared to surgeon B (please refer to “Elliptical excision force model parameters and objective performance analysis” subsection for more details).

In addition, Fig. 6 shows that “SA” and “SB” are well aligned along the abruptness axis (both surgeons appear on the lowest end of the “abruptness” feature). However, our model also captures the subtle difference in surgeon performance: “SA” lies higher along the “energy” feature, whereas “SB” landed higher along the steadiness axis. The former reflects the fact that surgeon A cut with larger forces, and the latter captures the lack of force modulation exhibited by surgeon B.

We added the missing details on surgeon experience in the Results section (paragraph 1).

7. Unclear whether the metrics can quantify subtle differences regarding the skill of two subjects.

Response: Our results do show that the model can capture relatively subtle differences in performance (as discussed above), much more so than time and motion studies would be capable of. However, we agree that the degree to which this is possible may require additional experimentation in future work.

8. Novel metrics are tailored to the elliptical excision task; unclear how similar metrics can be derived for other and maybe more complex tasks

Response: Our model specifically captures the temporal features that describe the task execution, hence can be used in the analysis of highly procedural tasks characterized by multiple distinct execution phases, e.g. suturing. We added the suggestions of using our model in other tasks to the Discussion section.

9. All in all, no convincing evidence for the validity of the proposed metrics

Response:

As mentioned above, the core contribution of this work is not to propose a new metric or set of metrics for skill assessment. Rather we aim to show that incision forces offer significant insight into the behaviours adopted by surgeons, and introduce a temporal model that reliably captures a large number of these underlying factors driving cutting behaviour. Importantly, this allows for quantitative analysis of the often subjective assessments and decisions made by raters. We strongly believe that our open-source tool and model will open up a number of future directions in terms of quantifying these subjective assessments. We have clarified this in the revised paper.

Other changes:

1. Renamed “Normalized force” metric to “Scaled force” to avoid ambiguity with normalized force data used in the analysis.
2. Renamed “jerkiness” feature to “abruptness” to avoid semantic opposition to (unrelated) “steadiness” feature.

Reviewers' comments:

Reviewer #2 (Remarks to the Author):

The reviewer would like to thank the authors for being responsive to the first review. Most of my comments were addressed, however, some open questions remain:

The application of the model to surgical skill assessment is still a weak part of the paper.

- Since the raters assess the videos merely subjectively, there is no reliable ground truth. This is in fact reflected by the ICC of 0.45, which is relatively low (some sources interpret the range 0.4 – 0.59 as fair [A], while others say everything below 0.5 is poor). The statement that the calculated ICC "confirms the validity of the assessment" (page 3) should therefore be corrected.

[A] Cicchetti, D. V. (1994). Guidelines, criteria, and rules of thumb for evaluating normed and standardized assessment instruments in psychology

 Koo, T. K., & Li, M. Y. (2016). A guideline of selecting and reporting intraclass correlation coefficients for reliability research

- Given the not-so-reliable assessments, the fact that the traditional force-based metrics do not show a monotonic relationship with these assessments does not necessarily imply that the force-based metrics are insufficient: the assessments itself could be the problem (i.e. unreliable).

- The authors infer novel force-based features from the proposed model parameters based on PCA. Here the reviewer cannot understand how it is possible to assign an interpretation to the novel axes PC1, PC2, PC3 as well as their orthogonal axes. How is it possible to conclude that the orthogonal axes correspond to the features abruptness, steadiness and energy?

- The authors emphasize their modeling of temporal structure. However, the generative model consists of only two states and can model only short time series with a duration of 4 seconds. Therefore, it seems likely that more sophisticated procedural tasks like suturing are too complex to be modeled in a similar manner. This limitation should also be reflected in the paper.

We thank the reviewers for their helpful comments which helped us to improve our revised paper. In summary, we introduced the following changes:

- Added missing details and clarifications regarding interpretation of features from PCA plots
- Corrected the interpretation of the ICC and clarified the validity of the assessment. We acknowledge that this ICC is low. This is a core motivation for our work - we provide quantitative metrics that can explain the reasons for disagreement and differing opinions.
- Commented on limitations of our model in modeling more complex tasks

Reviewer 2

1. The application of the model to surgical skill assessment is still a weak part of the paper.

Response: Our model is no less applicable to surgical skill assessment than any other force-based metric, and we show that parameters map to independent expert evaluators. The model offers a unique advantage over traditional force-based metrics by capturing the time-dependent structure of force measurements, which can be important in characterizing behavior in some procedural tasks, such as elliptical excision. However, we stress the point that there is no single agreed definition of what it means to be skillful and we do not propose a measure that purports to provide this. Rather, we show that the metrics derived from our sensor and model can explain the skill preferences of individual assessors.

2. Since the raters assess the videos merely subjectively, there is no reliable ground truth. This is in fact reflected by the ICC of 0.45, which is relatively low (some sources interpret the range 0.4 – 0.59 as fair [A], while others say everything below 0.5 is poor). The statement that the calculated ICC "confirms the validity of the assessment" (page 3) should therefore be corrected.

Response: Corrected. We changed our ICC interpretation from “fair” to “poor” and removed the original statement regarding the validity of the assessment. We further expand our view on the assessment validity in the response to the following comment below.

3. Given the not-so-reliable assessments, the fact that the traditional force-based metrics do not show a monotonic relationship with these assessments does not necessarily imply that the force-based metrics are insufficient: the assessments itself could be the problem (i.e. unreliable).

Response: We agree with the reviewer, that the reported assessment is not reliable for establishing the common criteria (or ground truth) for skill evaluation. Instead, we treat each expert evaluation as valid; after all, there are no meaningful ways in which we could rate or value one expert over another. Therefore, we analyzed student performances with respect to each expert in isolation (Fig. 6). We added this clarification in the “Subjective evaluation of the incision skills” section, the last paragraph.

In our work, we illustrate how our method enables a convenient decomposition of the force measurements into amplitude- and temporal-based features, and how these can bring additional insights into analysis when compared to other force-based metrics which disregard the temporal structure of data. To make this contribution clearer, we extended Fig. 5 with additional subfigures to make the described decomposition of force measurements and its application clearer to readers. In addition, we included an additional Supplementary Fig. 9 that illustrates the effect of model parameters on learned behavior. Please refer to the reworked “Elliptical excision force model parameters and behavior analysis” section for additional details.

4. The authors infer novel force-based features from the proposed model parameters based on PCA. Here the reviewer cannot understand how it is possible to assign an interpretation to the novel axes PC1, PC2, PC3 as well as their orthogonal axes. How is it possible to conclude that the orthogonal axes correspond to the features abruptness, steadiness and energy?

Response: We thank the reviewer for this excellent question! We substantially reworked the “Elliptical excision force model parameters and behavior analysis” section to explain this in detail (see the last three paragraphs). In addition, we added Supplementary Fig. 8 which illustrates the correlation between principal components and model parameters.

5. The authors emphasize their modeling of temporal structure. However, the generative model consists of only two states and can model only short time series with a duration of 4 seconds. Therefore, it seems likely that more sophisticated procedural tasks like suturing are too complex to be modeled in a similar manner. This limitation should also be reflected in the paper.

Response: We added the comment on applicability and limitations of our model in modeling procedural tasks with more complex data (“Discussion” section, the last paragraph).

Other changes:

1. Renamed the “steadiness” feature to “confidence”.

REVIEWERS' COMMENTS:

Reviewer #2 (Remarks to the Author):

The reviewer would like to thank the authors for being responsive to the open comments/questions. The current paper is an improved version, my open questions and concerns were carefully addressed. Overall, the authors provided sufficient adjustments that warrant a publication and I recommend it for publication.

REVIEWERS' COMMENTS:

Reviewer #2:

The reviewer would like to thank the authors for being responsive to the open comments/questions. The current paper is an improved version, my open questions and concerns were carefully addressed.

Overall, the authors provided sufficient adjustments that warrant a publication and I recommend it for publication.

AUTHOR RESPONSE:

We greatly thank the reviewers for their valuable feedback and good questions that helped us improve our paper.